# HMGB1/NF-κB Axis, IL-8, and Cuproptosis Contribute to Cisplatin-Induced Testicular Injury: Protective Potential Effect of Thymol

**DOI:** 10.3390/biom15111595

**Published:** 2025-11-14

**Authors:** Layla Alkharashi, Iman Hasan, Aliyah Almomen, Qamraa H. Alqahtani, Yasmen F. Mahran, Amul M. Badr, Reem T. Atawia, Awatif Binmughram, Rehab Ali, Nadrah Alamri, Amira M. Badr

**Affiliations:** 1Department of Pharmacology and Toxicology, College of Pharmacy, King Saud University, P.O. Box 2457, Riyadh 11451, Saudi Arabia; lalkharashi@ksu.edu.sa (L.A.); ihasan@ksu.edu.sa (I.H.); ghamad@ksu.edu.sa (Q.H.A.); aalmugram@ksu.edu.sa (A.B.); reali@ksu.edu.sa (R.A.); 2Department of Pharmaceutical Chemistry, College of Pharmacy, King Saud University, P.O. Box 2457, Riyadh 11451, Saudi Arabia; alalmomen@ksu.edu.sa; 3Department of Pharmacology and Toxicology, Faculty of Pharmacy, Ain Shams University, Cairo 11566, Egypt; reemtarekatawia@pharma.asu.edu.eg; 4Department of Medical Biochemistry and Molecular Biology, Faculty of Medicine, Cairo University, Cairo 12613, Egypt; amal.abdelrehem@kasralainy.edu.eg; 5Department of Pharmaceutical Sciences, College of Pharmacy, Southwestern Oklahoma State University, Weatherford, OK 7310196, USA; 6College of Pharmacy, King Saud University, P.O. Box 2457, Riyadh 11451, Saudi Arabia; nadsamri@gmail.com

**Keywords:** cisplatin, testes, HMGB1, cuproptosis, IL-8, thymol

## Abstract

Background: Cisplatin (CP) use is associated with testicular toxicity. Cuproptosis-related genes are associated with dysfunctional spermatogenesis. Additionally, the HMGB1/NF-κB axis has been involved in cuproptosis-mediated inflammation. The aim of the current study was to investigate the effect of CP toxicity on the HMGB1/NF-κB axis and cuproptosis in the rat testis. The effect of thymol was also explored. Methods: Four groups of male Wistar rats were used: control, thymol (60 mg/kg P.O. daily for 2 weeks), CP (8 mg/kg i.p single injection), and CP+thymol. Results: CP induced a significant decrease in serum testosterone and LH. CP-induced oxidative stress was evident by the modulation of oxidative stress markers. The expressions of IL-8, NF-κB, and HMGB1 were induced by CP treatment, accompanied by increased expression of cuproptosis genes, including SLC31A1, FDX1, and DLAT. On the other hand, thymol antagonized CP testicular injury. Thymol’s effect was associated with reduced expressions of IL-8, NF-κB, HMGB1, and cuproptosis markers. Conclusions: Collectively, this study provides evidence of the possible potential role of the HMGB1/NF-κB axis and cuproptosis in CP-induced testicular injury and illustrates the protective effects of thymol against testicular damage, which are attributed, at least in part, to blunting HMGB1 and cuproptosis-related genes expression.

## 1. Introduction

Cisplatin (CP) is a commonly used effective chemotherapeutic agent against numerous types of cancer, including lung, breast, ovarian, and cervical ones. It primarily functions by binding to DNA nucleotides, creating cross-links in the DNA strands that prevent DNA replication, resulting in cell cycle arrest [1]. Along with its effectiveness in destroying cancer cells, it can also potentially injure non-cancerous cells, including the testes. CP was found to induce adverse effects on testicular function that can significantly impair male fertility [2]. Induction of oxidative stress and inflammation is considered the central mechanism underlying CP-induced testicular toxicity [3].

It is well known that spermatogenesis is a tightly controlled process that aims at the production of sperm in adequate count and with specific characteristics for normal reproduction. CP can interfere with spermatogenesis, inducing apoptosis of germ cells, which ultimately results in a reduction in sperm quality and count or even a complete halt in sperm production (Kaku et al., 2022 [4]). It also disrupts Leydig cell function, leading to a reduction in testosterone production, the primary male sex hormone. Furthermore, CP was found to damage Sertoli cells and induce testicular atrophy [5]. The extent of CP-induced testicular injury is both dose- and duration-dependent, which can limit the therapeutic effectiveness of CP [2]. However, until now, no specific remedy has been identified for clinical application to mitigate this potential toxic effect.

As was mentioned earlier, CP can induce testicular inflammation. Nuclear factor ķappa B (NF-κB) is a known inflammatory mediator that induces the expression of many inflammatory cytokines [6]. Additionally, the nuclear protein, high mobility group box 1 (HMGB1), plays a central role in the inflammatory response. HMGB1 is produced as a reaction to cell damage and activates Toll-like receptor 4 (TLR-4), which, in turn, activates NF-κB [7]. The implication of the HMGB1/NF-κB axis in testicular toxicity was previously reported in several studies [8,9]. Moreover, the HMGB1/NF-κB axis was also found to be a critical component of CP-induced nephrotoxicity and ototoxicity [10,11]. However, the involvement of HMGB1 in CP-induced testicular injury has yet to be elucidated.

Furthermore, CP can cause significant metal disturbances that lead to adverse effects. Variations in the levels of iron, cobalt, copper (Cu), and zinc have been observed in relation to CP treatment [12]. Recently, cuproptosis has emerged as a new type of cell death in which Cu plays a crucial role. Cu is one of the essential trace elements in the human body. It is a critical component of the antioxidant enzyme superoxide dismutase (SOD) and cytochrome c oxidase. However, maintaining specific intracellular Cu concentrations is vital for proper physiological function (Yang et al., 2024 [13]). This balance is achieved through a complex interplay of Cu importer proteins, such as Solute Carrier Family 31 Member 1 (SLC31A1), and Cu exporter proteins, including ATPase Cu-transporting α (ATP7A) and β (ATP7B). Excessive intracellular Cu can disrupt the cellular environment, leading to toxic effects [14].

Ferredoxin 1 (FDX1) facilitates the reduction of Cu^2+^ to Cu^+^, which can bind to the lipoylated dihydrolipoamide S-acetyltransferase (DLAT), inducing its oligomerization. This process results in proteotoxic stress and cuproptosis [15]. Recent studies have proposed that the FDX1-DLAT axis serves as a mediating pathway for cuproptosis, particularly in the context of CP-sensitive organ toxicities. Furthermore, research has linked CP-induced acute kidney injury (AKI) cuproptosis through the FDX1-DLAT axis, alongside upregulation of Cu transporters. Notably, agents that attenuate cuproptosis genes, such as quercetin, have been shown to alleviate CP-induced acute kidney injury (AKI) [16]. In addition to its nephrotoxic effect, CP induces ototoxicity through a similar mechanism, particularly affecting the mitochondria-rich tissues of the cochlea. Changes in FDX1 and lipoylation pathway-protein levels have been reported after CP treatment in models of CP-induced cochlear toxicity, an effect that was alleviated by the administration of curcumin-loaded graphene oxide quantum dots [17].

Liu et al. (2022) highlighted the potential contribution of HMGB1 in mediating cuproptosis-induced inflammation [18]. In addition, Wu et al. provided evidence of the cross-talk between FDX1 and NF-κB1, and that NF-ķB subunit 1 is a direct regulator that binds to the FDX1 promoter [19]. It is worth mentioning that the role of cuproptosis in testicular injury remains relatively unexplored. Zhang et al. documented that excess Cu induces cuproptosis in mice testes; however, no data are available about the involvement of cuproptosis in chemotherapy-induced testicular toxicity [20].

Oxidative stress is central to CP-induced toxicities; accordingly, natural antioxidants are being considered promising candidates to ameliorate these potential adverse effects (Yalçın et al. 2024 [6]). Among these dietary antioxidants is thymol, a phenolic monoterpene with various pharmacological advantages, including hepatoprotective and nephroprotective effects [21,22]. Thymol also protects against testicular injury induced by various toxicants, such as hexachlorobenzene and imidacloprid [23,24]. A study conducted by our team demonstrated that thymol can protect against CP-induced testicular toxicity (Badr et al. 2025 [25]). However, the ability of thymol to inhibit HMGB1 expression and cuproptosis, as part of its pharmacological action, has not been previously explored.

Therefore, the present study aims to elucidate the potential involvement of HMGB1 signaling and cuproptosis in cisplatin-induced testicular toxicity, and to assess whether thymol mitigates these effects through modulation of these pathways.

## 2. Materials and Methods

### 2.1. Materials

CP was procured from EBEWE Pharma (GmbH. Nfg. KG, Unterach am Attersee, Australia), and thymol was obtained from Abcam (Cambridge, MA, USA). Antibody for HMGB1 was purchased from Fine Test, Wuhan, China; FNab013924, and that for glyceraldehyde-3-phosphate dehydrogenase (GAPDH, Cat. # ab8245) was sourced from Abcam Inc., Cambridge, MA, USA.

The remaining chemicals were of superior analytical quality. They were formulated following conventional protocols.

### 2.2. Methods

#### 2.2.1. Experimental Design

A total of thirty-two adult male Wistar albino rats (weighing 180–200 g) were sourced from the Animal House at King Saud University, College of Pharmacy in Saudi Arabia (SA). The animals were kept at a 12-h light/dark cycle and maintained under standard conditions of temperature of 23 ± 2 °C and an air-conditioned atmosphere, and they had free access to food and water. The experimental design was approved by the Research Ethics Committee at King Saud University with IACUC number (KSU-SE-25-18). The procedures of animal handling and treatment adhered to the guidelines of the Institutional Animal Care and Use Committee. Four groups (8 rats per group) of rats were divided as follows:•Group I (Control): Administered a thymol vehicle solution orally every day for 14 days and received an intraperitoneal (i.p.) saline injection on the seventh day.•Group II (T): Administered a daily oral dose of thymol (60 mg/kg body weight) mixed in a vehicle (1% Tween-80 in normal saline) through gavage for a duration of 14 days and were injected with saline (intraperitoneally) on the 7th day (Badr et al. 2025 [25]).•Group III (CP): The vehicle solution was administered orally for 14 days, and on the 7th day, a single i.p. dose of CP (8 mg/kg) was injected (Badr et al. 2025 [25]).•Group IV (CP+T): treated for 14 days with oral thymol (60 mg/kg BW), starting seven days before a single CP injection (8 mg/kg, i.p.) and continuing seven days after that for 7 days.

On day 15, carbon dioxide (CO_2_) was used to euthanize the rats. CO_2_ euthanasia was performed by placing the animal in an enclosed chamber, exposing it to a gradually increasing concentration of CO_2_ until it lost consciousness, following the AVMA guidelines. This usually takes 1–2 min, followed by decapitation and sample collection. The clear sera were prepared from collected blood samples by centrifuging them for 30 min at 3000 rpm and 4 °C. After excision of the two testes, they were directly weighed and washed, and the cauda epididymis was exposed and sliced. Afterward, these tissue samples were kept at a pH of 7.4 in 2 mL phosphate-buffered saline (PBS) and preserved at 4 °C. Also, dilution of sperm suspension was made using PBS (pH 7.2) at a 1:20 ratio. After collection of testes, they were spliced into two parts: the first part was prepared for histological and immunohistochemical analyses by fixing the parts in 10% neutral buffered formalin. The remaining sections of the tests were maintained for biochemical and molecular analysis at −80 °C. All researchers conducting the histopathological and molecular biology assays were blinded to the experimental design and were not provided with any information about the sample groups.

#### 2.2.2. Sperm Count and Morphology

A hemocytometer was utilized to determine the total sperm count. Viability of sperms and their overall numbers were estimated using a formerly reported standard equation [26]. A drop of semen was mixed with 1% aqueous eosin Y and gently stirred for 15 s, followed by the addition of a drop of 10% aqueous nigrosine and further mixing. Then, 10 µL of the prepared mixture was placed onto a clean glass slide and covered with a coverslip for microscopic examination.

In addition, sperm morphology was assessed using hematoxylin and eosin (H and E) staining.

#### 2.2.3. Serum Testosterone and Luteinizing Hormones (LH)

Enzyme-linked immunosorbent assay (ELISA) kits with high sensitivity were utilized to measure the serum levels of testosterone (Cat. # K76 21-100) and LH (Cat. # EK731211).

#### 2.2.4. Determination of Testicular Cu and Oxidative Stress and Inflammatory Biomarkers

Testicular concentration of Cu^+2^ was determined colorimetrically at 580 nm using assay kit (Cat. # E-BC-K3100) purchased from ElabScience (Wuhan, China). Testicular total antioxidant activities, SOD (Cat. # SD 25 21), and MDA (Cat. # SD 25 28) were studied by applying Biodiagnostics kits (Bio Diagnostics Comp. Giza, Egypt) in accordance with the manufacturer’s guidelines. Protein assay kit (Cat. # PC 0020) was obtained from Solarbio (Wuhan, China). Additionally, ELISA kits obtained from MyBiosource (San Diego, CA, USA) were utilized to evaluate inflammatory biomarkers NF-kB-P65 (Cat. # MBS250513) and IL-8; chemokine (C-X-C motif) ligand 1 (CXCL1); Cat. # MBS9141543. The FDX1 ELISA kit was a product of Solarbio (Wuhan, China), Cat. # MBS99100052. The ELISA kit for HMGB1 was that of CUSABIO BIOTECH, Houston, TX, USA (EELO47).

#### 2.2.5. Histopathological Examination

To prepare the testes samples for histological analysis, they were fixed in 10% formalin for 24 h. Subsequently, samples were fixed in paraffin, sectioned at 5 µm, and finally stained with H&E and investigated under a light microscope.

#### 2.2.6. Quantification of mRNA Expression of Cuproptosis-Related Genes

The expression levels of *FDX1*, *DLAT*, *and SLC31A1* genes, as cuproptosis-related markers, were analyzed using Real-Time Polymerase Chain Reaction RT-PCR. Frozen testicular samples are used for RNA extraction using the extraction kit (RNeasy Mini kit Cat. # 74104, Qiagen, Hilden, Germany, GmbH), followed by assessment of its concentration and purity with a NanoDrop spectrophotometer (Thermo Fisher Scientific Inc., Waltham, MA, USA). β-actin served as the internal reference gene for normalization. Only RNA samples showing an A260/A280 ratio between 1.8 and 2.0 were used for cDNA synthesis.

Quantitative RT-PCR (qPCR) was performed with a Quant Studio^®^ RT-PCR Instrument utilizing SYBR Green qPCR Master Mix. The reference gene used was β-actin, and gene expression was measured using the 2^-ΔΔCt method. The thermal cycling protocol started with a hold at 94 °C for 15 min, followed by 40 cycles consisting of denaturation at 94 °C for 15 s, annealing at the primer-specific temperatures to the primers (60 °C for thirty seconds), and extension at 72 °C for thirty seconds. (Table 1) for 1 min.

#### 2.2.7. Western Blotting and Protein Expression Analysis

Protein expressions were investigated using Western blotting, as detailed previously. Briefly, proteins were extracted and quantified as described in [29]. A total of 40 μg protein was subjected to SDS-PAGE (12%) followed by protein transfer onto PVDF membranes. After blocking with 5% non-fat dry milk for 1 h at room temperature, the membranes were probed with anti-HMGB1 (Cat. # FNab013924) and anti-GAPDH (Cat. # ab8245) at a dilution of 1:1000, incubated overnight at 4 °C in Phosphate-buffer saline with Tween20 (PBST buffer). Following washing, the membranes were probed with an HRP-conjugated anti-rabbit secondary antibody (1:5000) for 1 h at room temperature. Finally, protein bands were visualized using an ECL detection system, and images were acquired with the Image Quant LAS 4000 (The Lab World Group, Hudson, CO, USA), and normalized to the corresponding GAPDH signal.

#### 2.2.8. Statistical Assessment

GraphPad Prism software version 8 (GraphPad, San Diego, CA, USA) was selected for statistical assessment of the data. One-way ANOVA and Tukey–Kramer post hoc test was selected for multiple comparison. A *p*-value of less than 0.05 is considered statistically significant.

## 3. Results

### 3.1. Effect of CP and Thymol on Body and Testicular Weight

The results demonstrated that treatment with thymol (T) alone yielded outcomes similar to those of the control group for all measured parameters, indicating that there were no negative effects on the testicular function. In comparison, rats treated with CP exhibited a notable decrease in the body weight ratio when assessed against the control group. The control and the thymol groups showed an average increase of 16.2 ± 2.1 g, and 14.8 ± 2.2 g, respectively. While the CP group showed a decrease in body weight of about 37.4 ± 7.1, and for the CP treated with thymol (CP+T), the decrease in body weight was an average of 20.1 ± 7.8 g. Nevertheless, the concurrent administration of thymol significantly mitigated this decrease (Figure 1). On the other hand, no difference in the testicular weight/body weight ratio % was observed between any of the treated groups (Figure 1).

### 3.2. Effect of Thymol on Sperm Indices, Morphology, and Male Sex Hormones in CP-Treated Group

As illustrated in Figure 2, rats treated with CP showed a notable decrease in sperm count (A) and viability (B), along with the appearance of sperms exhibiting morphological defects (Figure 2C). Nevertheless, the simultaneous treatment with thymol significantly enhanced sperm count and viability, while also improving sperm morphology, with normal head and tail features (Figure 2).

As shown in Figure 3, serum reproductive hormone levels, including LH and testosterone, were significantly diminished with CP treatment compared to control rats. However, the levels of these hormones were significantly increased following the treatment of CP-exposed rats with thymol, compared to the CP group, and maintained at a level comparable to the control.

### 3.3. Thymol Modulates Testicular Oxidative Stress Biomarkers in CP-Treated Group

Figure 4 demonstrates the impacts of CP and thymol treatment on oxidative stress markers present in testicular tissue. CP treatment significantly decreased SOD activity and total antioxidants, while increasing MDA levels in comparison to the control group. Thymol treatment countered oxidative stress by significantly elevating SOD1 activity and total antioxidants, relative to the CP group, and significantly reduced MDA.

### 3.4. The Effect of CP and Thymol on the Expression of Testicular IL-8 (CXCL8), HMGB1, and NF-kB-p65

Our results showed that CP significantly upregulated proinflammatory cytokines IL-8 and NF-κB, a key transcription factor involved in inflammatory signaling, and disrupted HMGB1 expression (Figure 5). HMGB1 expression was measured with both ELISA kit and also an immunoblot was presented showing a significant increase in HMGB1 expression in CP group compared to the control one. Thymol treatment effectively normalized the expression of IL-8, NF-κB-P65, and HMGB1, suggesting its potential to suppress inflammation and protect testicular function.

### 3.5. CP Modulates the Testicular Cu^+2^ Level, FDX1 Protein Level and the Expression of Cuproptosis Genes; FDX1, DLAT, and SLC31A1

CP administration significantly upregulated the expression of FDX1 and DLAT as compared with control rats (Figure 6A,B). Additionally, *SLC31A1* was markedly altered, reflecting disrupted testicular function (Figure 6C). Additionally, CP induced a significant increase in testicular Cu^+2^ level, compared to the control group (Figure 6D), **and a significant increase in FDX1 protein level** (Figure 6). On the other hand, thymol treatment effectively reduced the expression levels of these genes, as compared with the CP group. Furthermore, it reduced FDX1 protein level significantly as compared to the CP group and reduced the Cu^+2^ concentration to levels non-significantly different from that of the control group. (Figure 6). Additionally, the gene expression analysis of *DLAT, FDX1,* and *SLC31A1* was performed using normal control and Testicular Germ Cell Tumor (TGCT) based on Genotype -Tissue Expression (GTEx) and the Cancer Genome Atlas (TCGA) databases using the Gene Expression Profiling Interactive Analysis 2 (GEPIA2) database (http://gepia2.cancer-pku.cn, accessed on 25 October 2025). Results showed no significant difference between normal and tumor tissues (Appendix A). These findings present cuporoptosis as a potential contributor to CP reproductive toxicity through gene regulation without a significant effect on testicular cancer pathogenesis.

### 3.6. Thymol Ameliorates CP-Induced Testicular Histological Changes

H and E staining of testicular tissue from CP-treated rats revealed marked structural damage, including degeneration of seminiferous tubules, vacuolization, and disorganization of germinal epithelium. Many seminiferous tubules showed sloughed germ cells and disrupted basement membranes, indicating severe testicular toxicity. In contrast, thymol-treated rats demonstrated notable histological improvement, with restoration of seminiferous tubule architecture, reduced vacuolization, and minimal interstitial changes (Figure 7). These findings suggest that thymol, as a candidate natural agent, possesses a role in preserving testicular structure against CP-induced damage.

## 4. Discussion

Cancer incidence is continually increasing all over the world, affecting people of various ages [30]. Due to improved treatment strategies, the survival rate of children and adults diagnosed with cancer has increased, with an increasing fraction experiencing post-cancer morbidities [31]. These morbidities include those linked to chemotherapy-induced toxicities, especially those affecting reproductive organs, and may lead to sexual dysfunction. The most affected are patients aged 18–39 [32]. CP is a highly effective chemotherapy; however, its use is associated with multiple toxic effects. Accordingly, there is growing interest in adjunctive therapies that have the potential to reduce CP-induced toxicities. Suitable remedies that can ameliorate its potential side effects can help increase effectiveness and tolerability [33]. One of the toxic effects of CP is testicular toxicity [34]. Understanding how toxic insults are triggered can help to create treatments that lessen their harmful effect while augmenting cytotoxic activity as well.

HMGB1 has been defined as one of the cancer prognostic markers that mediates tumor progression and resistance [35,36]. Furthermore, increased HMGB1 is also associated with the induction of inflammation in normal tissues [37]. Drugs that inhibit HMGB1 can help minimize resistance and adverse effects as well (Qiao et al. 2024 [38]; Liu et al. 2022 [18]). Additionally, cuproptosis represents a newly characterized Cu-mediated mechanism of cell death [20]. It is being investigated for its potential role in different toxin-induced cell injury models [39]. Nevertheless, the involvement of cuproptosis in chemotherapy and CP-induced testicular toxicity was not previously documented.

To fulfill our aim, CP was injected as a single dose. CP-induced testicular toxicity was confirmed initially by measuring sperm count and assessing sperm viability. Exposure to CP resulted in a marked reduction in sperm count, motility, and viability. This reduction was accompanied by a significant decline in testosterone and LH levels. Decreased LH may be explained in terms of CP’s effect on the pituitary gland [40]. The impact of CP on sperm count and male sex hormones is in accordance with previous studies [40,41]. On the other hand, CP didn’t affect the testicular weight/body weight ratio; this observation corroborates the findings of [42,43]. CP–induced testicular injury was further documented by histological examination using H and E, showing marked histological damage. We also previously examined that CP-induced testicular injury was associated with decreased Johnson’s score, and illustrated that CP’s effect on male sex hormones can be explained in terms of its effect on the mRNA expression of steroidogenesis enzymes [25].

As oxidative stress is a key player in various toxic insults, TAO, SOD, and MDA were assessed. CP-induced oxidative stress was evident by the reduction of TAO and SOD levels. Reduced antioxidant levels were accompanied by increased lipid peroxidation and increased MDA. CP is known to induce oxidative stress in different tissues ( [33]). Our results on CP-induced oxidative stress in testicular tissues are consistent with previously reported ones. Balanced ROS has been found to be important for the normal function of the spermatozoa, while increased ROS in the face of reduced antioxidants leads to oxidative stress [3,44]. The spermatozoa are highly vulnerable to oxidative stress due to their high polyunsaturated fatty acids in their membranes and limited antioxidant capacity [45]. Oxidative stress has been linked to reduced male fertility, with increased ROS in the seminal plasma of up to 80% of sub-fertile patients [46,47]. Increased ROS can substantially damage sperm membrane through lipid peroxidation, which adversely affects its shape, motility, and viability [48].

Oxidative stress is also known to be able to activate inflammatory pathways, explaining why inflammation contributes prominently to CP-induced injury (Elmorsy et al. 2024 [33]). HMGB1 is a known inflammatory mediator. HMGB1 is a member of the damage-associated molecular pattern (DAMP) mediators, which are produced in response to cellular injury. ROS are able to oxidize specific cysteine residues in HMGB1, which culminates in its release and activation [45]. Once activated, it can regulate immune response and activate inflammatory mediators, such as NF-κB [11]. This primarily occurs by binding to TLR-4 and the receptor for advanced glycation end-products (RAGE), resulting in NF-κB activation (Demir et al. 2024 [9]). NF-κB mediates the expression of many inflammatory cytokines and proapoptotic factors [7]. Thus, the HMGB1/NF-κB pathway encourages further synthesis of inflammatory mediators that contribute to testicular damage [7,9]. Additionally, inhibition of HMGB1 was shown to suppress oxidative stress; thus, cross-talk between oxidative stress and HMGB1 activation forms a positive feedback loop [38].

The HMGB1/NF-κB axis activation was previously documented to take part in testicular injury (Demir et al. 2023, 2024 [8,9]). It was also found to play a significant role in CP-induced nephro- and oto-toxicity (Fu et al. 2023 [11]; Qiao et al. 2024 [38]). Fu et al. provided evidence that CP induces the transcription of HMGB1 as well as its nuclear release in renal tubular cells. They also confirmed the role of HMGB1 in NF-κB activation, as silencing HMGB1 was associated with reduced NF-κB and inflammatory cytokines expression [11]. Similar results were observed in a CP-induced ototoxicity model by Qiao et al., 2024 [38]. However, the role of HMGB1 in CP-induced testicular injury was not previously studied. In the present study, CP induced the expression of both HMGB1 and NF-κB, demonstrating the upregulation of the HMGB1/NF-κB axis in CP-induced testicular inflammatory injury. The effect of CP on testicular NF-κB was previously reported [49].

Another remarkable inflammatory mediator is IL-8. IL8, together with IL-6, was found to be the main inflammatory mediator in the male genital tract, and its levels in seminal fluid are much greater than serum levels [50]. It plays a role in the chemoattraction of inflammatory cells and their activation. However, the function of IL-8 in the normal testes is still not clear [51]. Recently, a study highlighted increased IL-8 expression in Leydig cells in response to lipopolysaccharide-induced injury [52]. IL-8 transcription is promoted through NF-κB, in addition to other regulators [50]. Our data demonstrated that CP treatment was found to induce a significant increase in IL-8 in the testis, providing further insight into the mechanisms of CP-induced testicular damage.

As mentioned earlier, cuproptosis is a new kind of cell death caused by Cu (Shi et al. 2024 [39]). The aim of our research is to explore whether cuproptosis plays a role in CP-induced testicular injury. Different proteins contribute to maintaining the intracellular Cu level, one of which is SLC31A1, which mediates intracellular Cu influx. FDX1 reduces Cu^+2^ to Cu^+^, which has the ability to induce protein aggregation. Cu^+^ can bind to lipoylated DLAT, one of the constituents of the pyruvate dehydrogenase complex, causing it to aggregate and induce cell death. FDX1 plays a direct role in regulating the lipoacylation of proteins, including DLAT. Thus, FDX1 is crucial for cuproptosis induction (Zhang et al. 2024 [20]). Excess Cu^+^ can also be sequestered by binding to GSH. Accordingly, with increased oxidative stress and GSH depletion, the toxicity of free Cu^+^ increases [53].

In this study, CP’s potential to induce cuproptosis is documented. CP upregulated the expression of the SLC31A1 gene, which has a role in intracellular Cu influx. Moreover, CP increased FDX1 protein level, as well as DLAT gene expression, significantly as compared to the control. CP also induced a significant increase in testicular Cu level. These data provide evidence that cuproptosis may contribute to CP-induced testicular injury. The role of Cu in testicular injury was supported by previous data reporting that higher Cu levels in the semen and plasma are associated with decreased sperm count and motility, in addition to increased oxidative stress markers [14,54]. Cu in excess was also linked to the induction of inflammation. Cu was found to activate the TLR4/NF-κB inflammatory pathway [55]. Furthermore, in tumor cells, Cu was able to affect immune system regulation and increase the extracellular release of HMGB1 [52].

The involvement of cuproptosis in CP-induced toxicity was previously documented in CP-induced nephrotoxicity models. CP was found to be able to increase renal Cu^+^ concentration, and Cu^+^ supply amplified CP-induced cell death [16]. Increased Cu may be explained in terms of increased intracellular Cu intake or reduced Cu excretion [56]. CP upregulation of the SLC31A1 gene expression can explain its ability to increase Cu concentration and induce cuproptosis. This is supported by a recent study showing that the overexpression of SLC31A1 induced renal tubular cell cuproptosis, which was inhibited by Cu chelators [16]. Moreover, CP treatment in our study was associated with increased expression of FDX1 and DLAT, that together orchestrate the pathway of cuproptosis. Collectively, this study provides the first evidence linking cuproptosis-related gene dysregulation to cisplatin-induced testicular injury.

The thymol effect on CP-induced testicular toxicity was previously documented (Badr et al. 2025 [25]). The work seeks to elucidate thymol’s mechanism of action further. Thymol exhibited antioxidant activity as evidenced by reduced lipid peroxidation and increased total antioxidants and SOD. This was in harmony with previous studies across various tissues [22,24]. Additionally, thymol was found to inhibit inflammation by downregulating the HMGB1/NF-κB axis. It induced a significant decrease in HMGB1 and NF-κB levels. This can be explained in terms of oxygen scavenging ability and reduced oxidative stress. The effect of thymol on NF-κB has been previously documented [57,58]. Nonetheless, the suppressive effect of thymol on HMGB1 was not studied before. Furthermore, thymol inhibited the expression of IL-8 [59], further supporting its anti-inflammatory activity.

Regarding cuproptosis, thymol showed potential inhibitory activity. It inhibited the gene expression of SLC31A1 and DLAT and reduced the FDX1 protein levels. By inhibiting SLC31A1, thymol has the potential to reduce intracellular Cu influx. This was further confirmed by a significant reduction of testicular Cu level in the CP group treated with thymol. Thymol treatment also reduced FDX1 levels, thus inhibiting the formation of the more reactive Cu^+^. Furthermore, through its powerful antioxidant activity, thymol can directly scavenge hydroxyl radicals (OH^−^) and other ROS formed as a result of Cu^+^ reaction with H_2_O_2,_ and other molecules, protecting cells from lipid peroxidation and DNA damage, which finally leads to cell injury and death [60,61].

The outcome of thymol on the expression of cuproptosis genes and proteins has not been previously explored. Moreover, the TCGA-TGCT expression data analysis suggests that cuproptosis does not significantly affect testicular tumor progression, unlike other cancer types, or it might be stage-dependent. Thus, the protective effects of thymol on testes are unlikely to interfere with the anticancer effects [62]. This is the first study, to our knowledge, to explore the thymol cuproptosis-inhibitory effect. The preservative properties of thymol against CP-induced testicular injury were confirmed by histological examination and its effect on male sex hormones, supporting data from [25].

## 5. Conclusions

In conclusion, this study confirms that induction of oxidative stress and inflammation are key players in CP-induced testicular injury. It also documents CP-mediated upregulation of HMGB1/NF-κB, IL-8, and cuproptosis markers in testicular tissues, shedding light on their potential role in testicular injury. Additionally, the gonadoprotective activity of thymol was further supported, and the inhibition of HMGB1, as well as cuproptosis, is documented as a new potential protective mechanism. As a future direction, the effect of cuproptosis in CP-induced testicular injury can be further documented by examining its impact on the protein expression of DLAT, LIAS, and other cuproptosis markers. In addition, we recommend using a specific cuproptosis inhibitor or knocking down cuproptosis genes to validate further the role of cuproptosis in the CP testicular injury model.

## Figures and Tables

**Figure 1 biomolecules-15-01595-f001:**
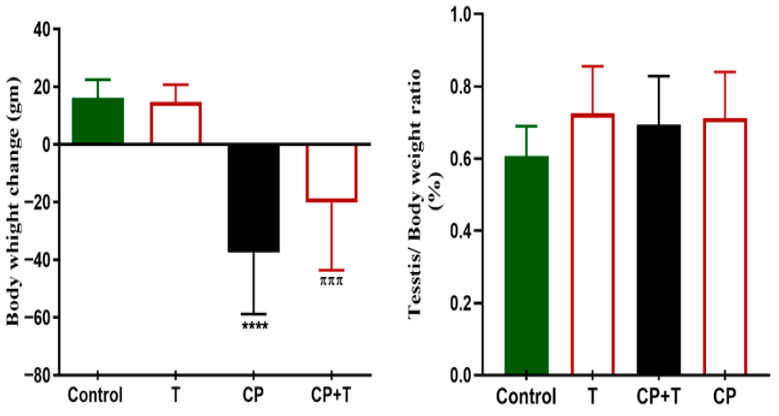
Body Weight and Testis Weight/Body Weight change in all Treated Groups. The variations in body and testis weight/body weight between the CP and CP+T groups were plotted as mean ± SEM, using ANOVA followed by Tukey–Kramer as a post hoc test. T: thymol; CP: cisplatin; CP+T: cisplatin and thymol. **** (*p* < 0.001): as compared to control. ^πππ^ (*p* < 0.001): as compared to CP (*n* = 8).

**Figure 2 biomolecules-15-01595-f002:**
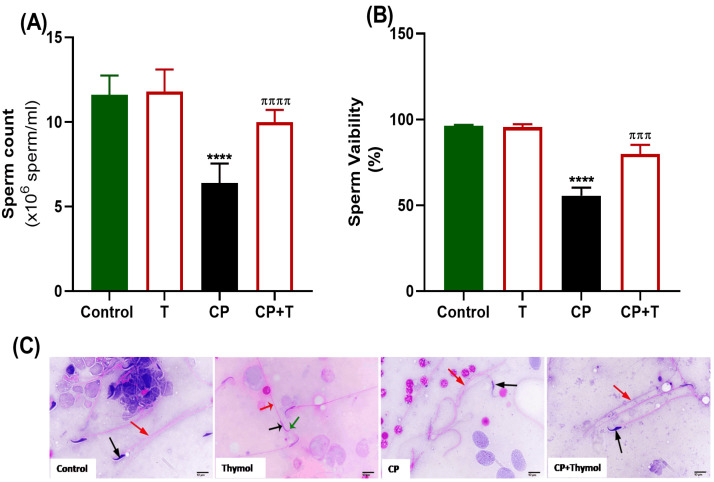
Effect of Thymol on Sperm Indices and Morphology. (**A**) Impact of thymol on total sperm count and (**B**) viability % in both control and CP-treated groups. (**C**) Sperm morphology. A photomicrograph of sperm from the control group illustrates a normal head (black arrow) and a straight tail (red arrow). Photomicrograph of epididymal sperm from the Thymol-group shows healthy rat sperm with a head (black arrow), hook (green arrow), and straight tail (red arrow). Photomicrograph of the CP group displays a shrunken head (black arrow) associated with a bent tail (red arrow). Photomicrograph of animal sperm from the CP+Thymol group reveals a normal head (black arrow) with a normal straight tail (red arrow) (H&E-400X). T: thymol; CP: cisplatin; CP+T: cisplatin and thymol. **** (*p* < 0.001, and 0.0001, respectively): as compared to control. ^πππ^ (*p* < 0.001) and ^ππππ^ (*p* < 0.0001): as compared to CP. n = 6.

**Figure 3 biomolecules-15-01595-f003:**
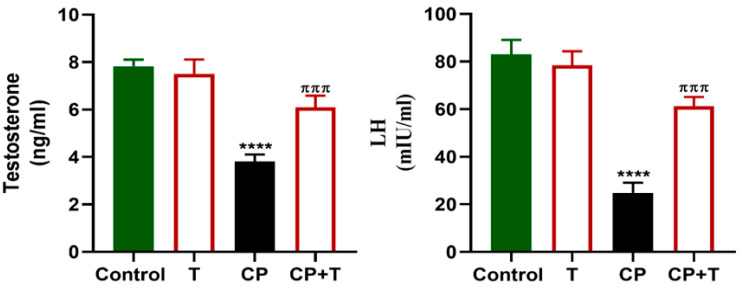
Thymol Elevates Testosterone and LH levels in CP-Induced Testicular Toxicity in Rats. Testosterone and LH levels in the serum following cisplatin and cisplatin+ thymol treatment were demonstrated as mean ± SEM, using ANOVA followed by Tukey–Kramer as a post hoc test. LH: lutienizing hormone, T: thymol; CP: cisplatin; CP+T: cisplatin and thymol. **** (*p* < 0.001): as compared to control. ^πππ^ (*p* < 0.001): as compared to CP. n = 6.

**Figure 4 biomolecules-15-01595-f004:**
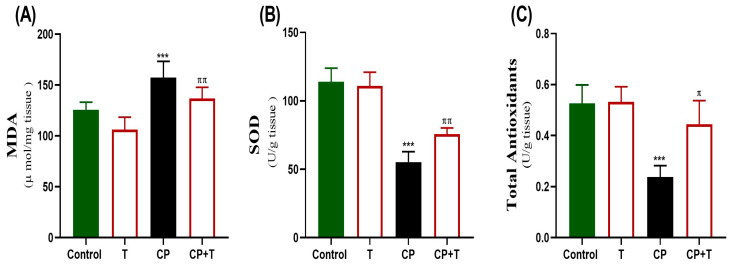
The protective effect of thymol on CP-induced testicular oxidative stress. The levels of MDA (**A**), SOD (**B**), and Total antioxidant levels (**C**) activity in the testicular tissue following cisplatin and CP+ thymol treatment were demonstrated as mean ± SEM, using ANOVA followed by Tukey–Kramer as a post hoc test. MDA: Malondialdehyde, SOD: superoxide dismutase, T: thymol; CP: cisplatin; CP+T: cisplatin and thymol. *** (*p* < 0.001): as compared to control. ^π^ (*p* < 0.05) ^ππ^ (*p* < 0.01): as compared to CP. *n* = 6.

**Figure 5 biomolecules-15-01595-f005:**
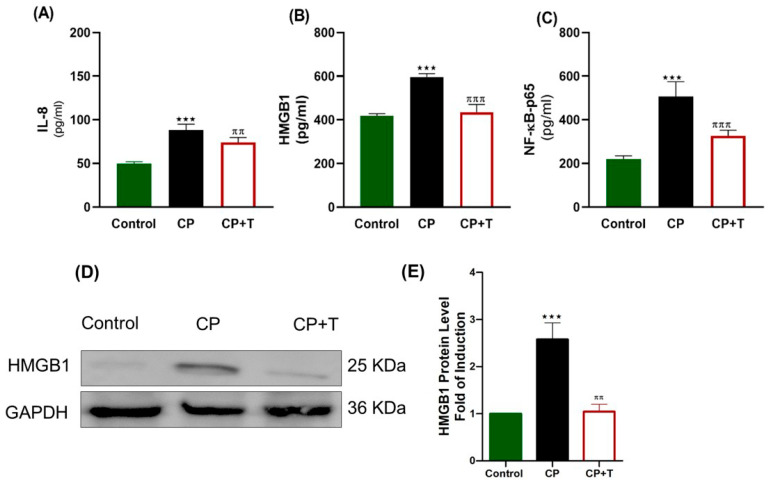
CP Toxicity Upregulates Testicular IL-8, HMGB1, and NF-kB-P65 Expression in Rats. Proinflammatory markers were measured using specific ELISA kits, and the results were plotted as mean ± SEM (*n* = 8). (**A**) IL-8, (**B**) HMGB1, and (**C**) NF-κB-p65. *n* = 8. Western Blotting Analysis of HMGB1 was presented in (**D**), and quantitative expression of Western blot data was presented as (**E**), n = 3. The differences between groups were analyzed using ANOVA followed by Tukey–Kramer as a post hoc test. IL: interleukin, CP: cisplatin; CP+T: cisplatin and thymol, IL-8: interleukin-8. *** (*p* < 0.001): CP compared to control, ^ππ^ (*p* < 0.05) and ^πππ^ (*p* < 0.001): as compared to CP.

**Figure 6 biomolecules-15-01595-f006:**
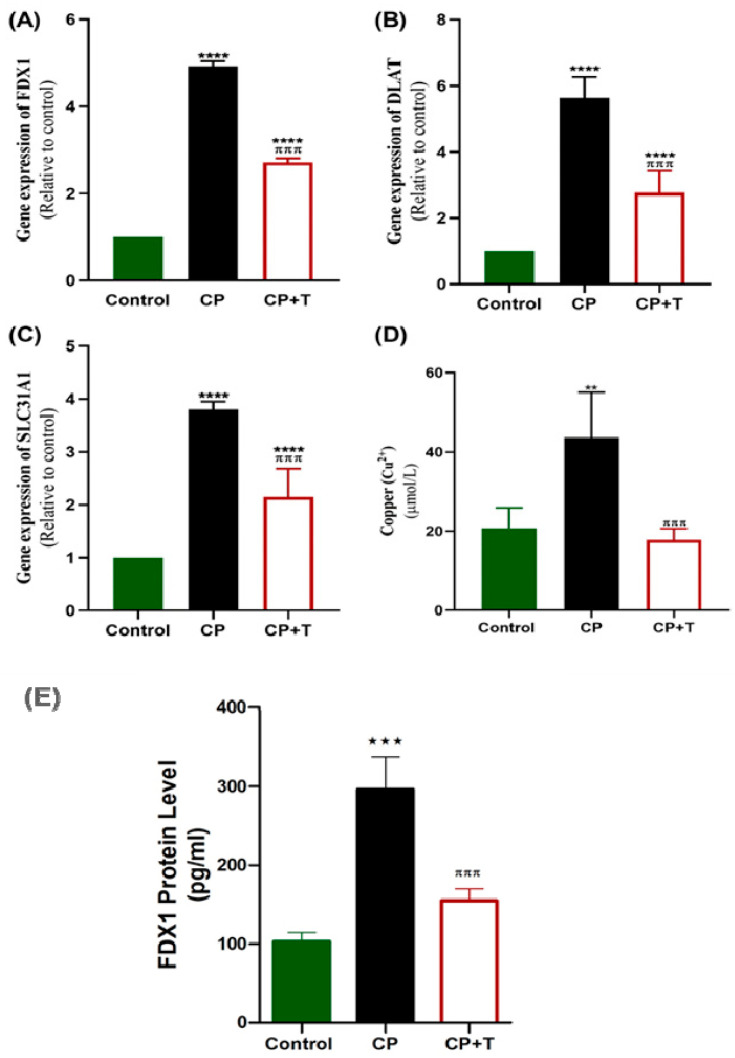
CP Upregulated Testicular Gene Expression of *FDx1, DLAT,* and *SLC31A1* and Increased Testicular Copper and FDX1 Protein in Rats. RT-PCR was used to assess gene expression of *FDx1* (**A**), *DLAT* (**B**), and *SLC31A1* (**C**), in CP-induced testicular toxicity in rats with and without thymol treatment. Copper (Cu ^2+^) was also assessed (**D**). (**E**) The effect on FDX1 protein level. The results are presented as mean ± SEM (*n* = 8). CP: cisplatin; CP+T: cisplatin and thymol. (** *p* < 0.01, *** *p* < 0.001 **** *p* < 0.0001: CP compared to control & ^πππ^ *p* < 0.001: CP+T compared to CP).

**Figure 7 biomolecules-15-01595-f007:**
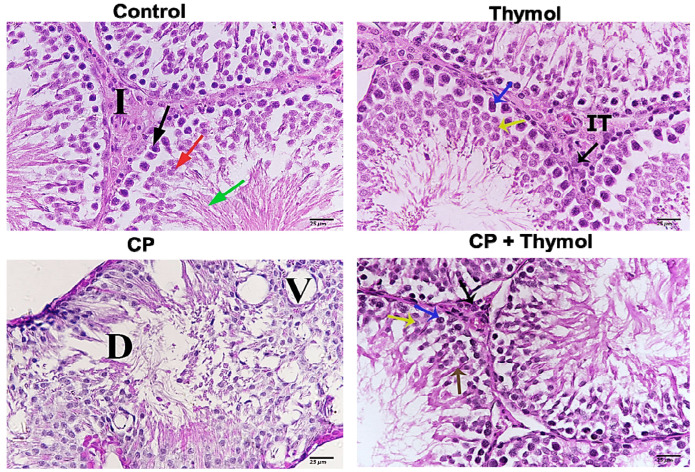
Thymol preserved the testicular structure and spermatogenesis against CP-induced damage H and E was performed to study the impact of thymol on CP-induced testicular damage. Representative sample from the control group showing seminiferous tubules filled with different stages, Leydig cells (black arrow), secondary spermatocytes (red arrow), spermatozoa (green arrow), and interstitial tissue (I). Testicular tissue from the thymol-treated group, showing normal histological features; spermatocyte A (blue arrow), and spermatocyte B (yellow arrow). CP-treated rats revealed marked structural damage, degeneration in the seminiferous tubule (D), and vascular degeneration (V). A sample of CP + thymol shows Leydig cells (black arrow), spermatozoa (brown arrow), spermatocyte A (blue arrow), and spermatocyte B (yellow arrow). CP+ thymol-treated rats demonstrated notable histological improvement. X-400. CP: Cisplatin.

**Table 1 biomolecules-15-01595-t001:** Primer Sequences.

Gene Name	Primer Sequences	Ref.
*FDX1*	CAAGGGGAAAATTGGCGACTC TTGGTCAGACAAACTTGGCAG	NM_007996 [20]
*DLAT*	TCCCTCCGCATCAGAAGGTT CCAACTGGAACATCTCTGGTC	NM_145614 [20]
*SLC31A1*	TATTTGGTGGCTGGGGTTCT CACTAGGTCTGGAGAGGCAC	NM_133600.3 (El-Nablaway et al. 2025 [27])
*β- actin*	CCTGCTTGCTGATCCACA CTGACCGAGCGTGGCTAG	[28]

## Data Availability

The original contributions presented in this study are included in the article/Appendix A. Further inquiries can be directed to the corresponding author.

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
