# Peer review of "HMGB1/NF-κB Axis, IL-8, and Cuproptosis Contribute to Cisplatin-Induced Testicular Injury: Protective Potential Effect of Thymol"

_biomolecules, 2025, doi:10.3390/biom15111595_

Round 1

Reviewer 1 Report

Comments and Suggestions for Authors

 This manuscript of  Alkharashi et al. addresses an important and timely topic by investigating the protective effects of thymol against cisplatin-induced testicular toxicity, with a focus on oxidative stress, inflammatory signaling via HMGB1/NF-κB, and the emerging mechanism of cuproptosis. The study is well designed, employing a comprehensive multimodal approach that includes biochemical, molecular, and histopathological analyses, which together provide robust support for the conclusions.

The work contributes novel insights by linking cuproptosis-related pathways to cisplatin-induced gonadotoxicity, a relatively unexplored area, and by proposing thymol as a potential modulator of these mechanisms. The manuscript is generally well written and organized, with clear presentation of results and thoughtful discussion.

However, to strengthen the manuscript’s scientific rigor and clarity, I recommend revisions focused on improving the logical flow in the Introduction, expanding methodological details to ensure reproducibility, and providing more precise quantitative data and statistical reporting in the Results. Additionally, the Discussion would benefit from a clearer distinction between previously established findings and novel contributions, as well as a candid acknowledgment of study limitations.

With these improvements, the manuscript will represent a valuable contribution to the field of reproductive toxicology and chemoprotective interventions.

Introduction

Overall Evaluation
The introduction is generally well-written and logically structured, providing an effective balance between general background and the specific rationale of the study. The authors clearly describe cisplatin (CP)-induced testicular toxicity and the associated molecular mechanisms, including oxidative stress, inflammation through the HMGB1/NF-κB axis, and the emerging concept of cuproptosis. However, several refinements could further improve clarity, logical flow, and scientific precision.

Major Comments

  • Logical Coherence:
    The transition from CP-induced toxicity to the concept of cuproptosis is abrupt. Consider adding a bridging statement explaining why cuproptosis might plausibly contribute to CP-induced testicular injury (e.g., through metal ion imbalance or mitochondrial disruption).
  • Mechanistic Detail:
    The description of the FDX1–DLAT–proteotoxicity pathway is concise but could be expanded to mention whether this mechanism has been reported in other CP-sensitive organs such as the kidney or cochlea. This would enhance the biological plausibility of the hypothesis.
  • Citations:
    Some references appear redundant (e.g., repeated citations of the same author/year). Consolidating or reorganizing these citations would improve readability.
  • Study Aim:
    The final aim statement could be rewritten for precision and stylistic alignment with Biomolecules standards:
    “The present study aims to elucidate the potential involvement of HMGB1 signaling and cuproptosis in cisplatin-induced testicular toxicity, and to assess whether thymol mitigates these effects through modulation of these pathways.”
  • Language and Style:
    Minor grammatical and typographical errors should be corrected (e.g., “is worth to mention” → “it is worth mentioning”). A careful language edit is recommended to enhance fluency and professionalism.

  1. Materials and Methods

Overall Evaluation
The Materials and Methods section is comprehensive and generally appropriate for the study’s objectives. The experimental design, ethical approval, and multimodal approach (biochemical, molecular, and histological) are clear strengths. However, additional methodological details and clarifications are needed to ensure full reproducibility and alignment with international reporting standards (ARRIVE and MIQE guidelines).

Points Requiring Clarification

  • Animal Model and Sample Size:
    Include whether a power analysis was performed to justify the sample size (n = 8). Indicate whether randomization and blinding were used during treatment and data analysis.
  • Dosing Justification:
    Provide a brief rationale for the CP (8 mg/kg, i.p.) and thymol (60 mg/kg, oral) doses, citing previous studies or pilot data confirming their efficacy and safety.
  • Animal Husbandry and Euthanasia:
    Report precise environmental conditions (temperature, humidity, and light/dark cycle).
    Confirm compliance of the euthanasia method (CO₂ asphyxiation followed by decapitation) with AVMA 2020 guidelines.
  • Sample Collection:
    Clarify whether both testes were used for analysis and how samples were distributed across histological, biochemical, and molecular assays.
    Verify the tissue storage temperatures—“37 °C” likely represents a typographical error (should be 4 °C).
  • Biochemical and Molecular Assays:
    Provide catalog numbers, manufacturers, and normalization methods for biochemical assays (e.g., per mg protein).
    Indicate whether assays were performed in duplicate or triplicate.
    For RT-qPCR, report primer efficiencies, annealing temperatures, and inclusion of negative controls.
    For Western blotting, specify the gel percentage, blocking conditions, antibody sources/dilutions, and detection system.
  • Statistical Analysis:
    Indicate whether data normality (Shapiro–Wilk test) and variance homogeneity (Levene’s test) were confirmed before ANOVA.
    Clarify outlier handling, number of replicates, and exact significance thresholds (e.g., p < 0.05, p < 0.01).
  • Formatting and Consistency:
    Standardize gene and protein nomenclature (e.g., SLC31A1, not SLC3A1).
    Ensure consistent formatting of units, degree symbols, and chemical notation (Cu²⁺).

  1. Results

Overall Evaluation
The results are logically structured and effectively demonstrate the protective effect of thymol against CP-induced testicular toxicity through oxidative, inflammatory, and cuproptotic mechanisms. However, numerical transparency, figure annotations, and methodological details can be improved.

Key Comments

  • Body and Testicular Weights:
    Define whether “body weight ratio” refers to testis/body-weight ratio or percentage body-weight change. Include numerical data (mean ± SEM) in the text or supplementary materials.
  • Sperm Parameters and Hormones:
    Provide quantitative data for sperm morphology and viability percentages.
    Specify the viability staining method used (e.g., eosin–nigrosin).
    Indicate hormone concentration units and assay variability.
    Add sample size (n) and exact p-values in figure legends.
  • Oxidative Stress Markers:
    Clarify normalization (per mg protein or g tissue).
    Ensure consistency of units (e.g., nmol MDA/g tissue).
    Consider visual representation (bar graphs) in addition to tables.
  • Inflammatory Markers (HMGB1, NF-κB, IL-8):
    Report ELISA results in consistent units.
    Clearly separate Western blot (protein expression) and ELISA (cytokine levels) data.
    Include representative blots labeled with molecular weights.
  • Cuproptosis Gene Expression:
    Provide fold-change values with SEM from 2^-ΔΔCt analysis.
    Confirm reference gene stability (β-actin).
    Ensure consistent gene nomenclature.
    Discuss correlations between cuproptosis and oxidative/inflammatory parameters.
  • Histopathology:
    Add semi-quantitative scoring (e.g., Johnsen’s score) to support descriptive findings.
    Include scale bars and consistent magnifications.
    Confirm that histological assessment was blinded.

  1. Discussion and Conclusions

Overall Evaluation
The Discussion effectively integrates the study’s biochemical, molecular, and histological results, emphasizing the novel involvement of cuproptosis and the protective role of thymol. The logical flow is strong, but the section could be more concise, with clearer delineation between established knowledge and new contributions.

Major Comments

  • Contextualization:
    The initial discussion repeats background information already presented in the Introduction. Consider condensing to focus on new insights.
    Rephrase statements like “claims to minimize its use” to more neutral scientific language (e.g., “There is growing interest in adjunctive therapies that reduce cisplatin-induced toxicity…”).
  • Mechanistic Insights:
    Expand on how CP triggers HMGB1 release (e.g., via DNA damage or mitochondrial dysfunction).
    Clarify the sequence between oxidative stress and HMGB1/NF-κB activation.
    Specify whether IL-8 elevation originates primarily in Leydig cells or seminiferous epithelium.
    Discuss how CP may interfere with copper metabolism, thus activating cuproptosis.
    Highlight novelty:
    “This study provides the first evidence linking cuproptosis-related gene dysregulation to cisplatin-induced testicular injury.”
  • Thymol Mechanisms:
    Summarize thymol’s multi-target effects:
    • Antioxidant: Increases SOD, reduces MDA.
    • Anti-inflammatory: Suppresses HMGB1/NF-κB/IL-8 signaling.
    • Anti-cuproptotic: Downregulates FDX1, DLAT, and SLC31A1.
      Clarify whether these effects stem from direct copper modulation or secondary antioxidant activity.

The discussion successfully synthesizes the biochemical, molecular, and histological findings, particularly highlighting the novel involvement of cuproptosis and the protective role of thymol in cisplatin-induced testicular injury. To further strengthen the contextual framework and emphasize the study’s novelty, I suggest incorporating recent relevant literature.

First, the study by Berman et et al. (2025), DOI: 10.1016/j.cbi.2025.111747 “Comprehensive characterization of poly(ADP-ribosyl)ation in spermatozoa as a novel and early biomarker of sperm health: A preliminary look,” provides valuable insights into early biomarkers of sperm health through poly(ADP-ribosyl)ation profiling. Including this reference would enrich the discussion of molecular markers linked to sperm integrity and damage induced by cisplatin, especially in relation to DNA damage and oxidative stress pathways.

Second, the recent work on nano Spirulina platensis (NSP) by Khalil et al. (2024), DOI: 10.1007/s00210-024-03483-z “Nano Spirulina platensis countered cisplatin-induced repro-toxicity by reversing the expression of altered steroid hormones and downregulation of the StAR gene,” highlights another potential therapeutic avenue. NSP’s ability to restore steroid hormone balance and normalize StAR gene expression complements the current findings by emphasizing the importance of hormonal regulation and steroidogenesis in mitigating cisplatin toxicity. This adds a broader perspective on potential multi-target protective strategies that include antioxidant, anti-inflammatory, and endocrine modulatory effects.

Incorporating these references will:

  • Strengthen the argument that early molecular alterations (e.g., poly(ADP-ribosyl)ation) serve as sensitive indicators of sperm health, consistent with the observed HMGB1-mediated damage and oxidative stress.
  • Broaden the discussion on therapeutic interventions beyond thymol, pointing to complementary or synergistic strategies such as NSP for preserving fertility during chemotherapy.
  • Emphasize the multifactorial nature of cisplatin-induced testicular injury, involving oxidative stress, inflammation, steroidogenic disruption, and newly identified cell death pathways like cuproptosis.

Technical Corrections:

  • Use “H&E” consistently.
  • Correct citation formatting (e.g., “Kohsaka et al. (2020)” instead of “the finding of (Kohsaka et al. 2020)”).
  • Replace informal connectors (“Moreover”) with more formal alternatives (“Furthermore”).

Summary:
The manuscript presents novel and significant findings on the involvement of HMGB1 and cuproptosis in CP-induced testicular toxicity and the protective role of thymol. To further strengthen the work, the authors should (1) refine transitions and mechanistic rationale in the Introduction, (2) enhance methodological transparency, (3) provide more detailed data presentation and figure labeling in the Results, and (4) focus the Discussion on mechanistic interpretation and study implications rather than repetition of results.

Author Response

This manuscript of  Alkharashi et al. addresses an important and timely topic by investigating the protective effects of thymol against cisplatin-induced testicular toxicity, with a focus on oxidative stress, inflammatory signaling via HMGB1/NF-κB, and the emerging mechanism of cuproptosis. The study is well designed, employing a comprehensive multimodal approach that includes biochemical, molecular, and histopathological analyses, which together provide robust support for the conclusions.

The work contributes novel insights by linking cuproptosis-related pathways to cisplatin-induced gonadotoxicity, a relatively unexplored area, and by proposing thymol as a potential modulator of these mechanisms. The manuscript is generally well written and organized, with clear presentation of results and thoughtful discussion.

However, to strengthen the manuscript’s scientific rigor and clarity, I recommend revisions focused on improving the logical flow in the Introduction, expanding methodological details to ensure reproducibility, and providing more precise quantitative data and statistical reporting in the Results. Additionally, the Discussion would benefit from a clearer distinction between previously established findings and novel contributions, as well as a candid acknowledgment of study limitations.

With these improvements, the manuscript will represent a valuable contribution to the field of reproductive toxicology and chemoprotective interventions.

Introduction

Overall Evaluation

The introduction is generally well-written and logically structured, providing an effective balance between general background and the specific rationale of the study. The authors clearly describe cisplatin (CP)-induced testicular toxicity and the associated molecular mechanisms, including oxidative stress, inflammation through the HMGB1/NF-κB axis, and the emerging concept of cuproptosis. However, several refinements could further improve clarity, logical flow, and scientific precision.

Major Comments

  • Logical Coherence:

The transition from CP-induced toxicity to the concept of cuproptosis is abrupt. Consider adding a bridging statement explaining why cuproptosis might plausibly contribute to CP-induced testicular injury (e.g., through metal ion imbalance or mitochondrial disruption).

Response: Thank you for your valid comment. In response to that, we have added the following linking statement to the text in the suggested position: “In addition, CP can mediate remarkable metal disturbances, which contribute to its adverse effects. Changes in the levels of iron, cobalt, copper, and zinc have been found to be related to CP treatment, with relevant references.

  • Mechanistic Detail:

The description of the FDX1–DLAT–proteotoxicity pathway is concise but could be expanded to mention whether this mechanism has been reported in other CP-sensitive organs, such as the kidney or cochlea. This would enhance the biological plausibility of the hypothesis.

Response: We thank the reviewer for this valuable comment. The issue has been fully addressed in the revised manuscript. We have also highlighted this pathway in the graphical abstract.

  • Citations:
    Some references appear redundant (e.g., repeated citations of the same author/year). Consolidating or reorganizing these citations would improve readability.

Response: References were corrected. Thank you for your comment

  • Study Aim:

The final aim statement could be rewritten for precision and stylistic alignment with Biomolecules standards:
“The present study aims to elucidate the potential involvement of HMGB1 signaling and cuproptosis in cisplatin-induced testicular toxicity, and to assess whether thymol mitigates these effects through modulation of these pathways.”

Response: Done. We thank the reviewer for notifying us of this issue. The aim statement was revised and written in alignment with Biomolecules standards.

  • Language and Style:
    Minor grammatical and typographical errors should be corrected (e.g., “is worth to mention” → “it is worth mentioning”). A careful language edit is recommended to enhance fluency and professionalism.

Response: Thank you for your insightful comment. The manuscript has undergone careful language edit to enhance fluency and professionalism.

  1. Materials and Methods

Overall Evaluation

The Materials and Methods section is comprehensive and generally appropriate for the study’s objectives. The experimental design, ethical approval, and multimodal approach (biochemical, molecular, and histological) are clear strengths. However, additional methodological details and clarifications are needed to ensure full reproducibility and alignment with international reporting standards (ARRIVE and MIQE guidelines).

 Points Requiring Clarification

  • Animal Model and Sample Size:
    Include whether a power analysis was performed to justify the sample size (n = 8). Indicate whether randomization and blinding were used during treatment and data analysis.

Response:

Thank you for your constructive comment. A priori power analysis was performed using GraphPad Prism-8 to justify the sample size. Based on previously published data, the analysis indicated that at least 8 animals per group were sufficient to detect significant differences, given the risk of mortality. This has now been clarified in the methods section. Regarding randomization and blinding: Animals were randomly assigned to the experimental group; data collection and analysis were performed by an investigator blinded to the treatment groups to avoid bias. These details have been added to the manuscript.

  • Dosing Justification:
    Provide a brief rationale for the CP (8 mg/kg, i.p.) and thymol (60 mg/kg, oral) doses, citing previous studies or pilot data confirming their efficacy and safety.

Response:

Thank you for your insightful comment. The doses of cisplatin (CP) and thymol used in this study were selected based on previously published studies that demonstrated their efficacy and safety in similar experimental models. Specifically, cisplatin at a dose of 8 mg/kg (i.p.) has been widely employed to induce reproducible testicular and systemic toxicity without causing excessive mortality in rodents. Likewise, thymol at 60 mg/kg (oral) has been reported to exert significant antioxidant, anti-inflammatory, and cytoprotective activities with a favorable safety profile.

Importantly, the selection was also based on studies in our lab, which proved the dose effectively induced testicular injury. The same cisplatin dose was used in our previous study, where co-administration of a natural antioxidant (Morin hydrate) successfully ameliorated cisplatin-induced testicular damage without adverse effects or toxicity. In that work, the 8 mg/kg cisplatin dose reliably induced oxidative stress, ferroptosis activation, and dysregulation of steroidogenesis pathways, providing a well-established model of testicular injury suitable for evaluating protective agents. The antioxidant dose used in that study likewise showed strong protective effects without toxicity. This prior evidence supports the validity and reproducibility of the dosing protocol employed in the current study involving thymol. Also thymol was previously used by our team in a model of radiation-induced ovarian failure and fluorouracil-induced toxicity in the same dose, and showed protective effect with no toxic manifestations.

Accordingly, the same dosing scheme was adopted here to allow a consistent comparison of protective efficacy across studies and to maintain alignment with established experimental protocols.

Selected reference:

Kelles, M., Tan, M., Kalcioglu, M.T. et al. The Protective Effect of Chrysin Against Cisplatin İnduced Ototoxicity in Rats. Indian J Otolaryngol Head Neck Surg 66, 369–374 (2014). https://doi.org/10.1007/s12070-013-0695-x

Soni, K. K., Kim, H. K., Choi, B. R., Karna, K. K., You, J. H., Cha, J. S., … Park, J. K. (2016). Dose-dependent effects of cisplatin on the severity of testicular injury in Sprague Dawley rats: reactive oxygen species and endoplasmic reticulum stress. "Drug Design, Development and Therapy10, 3959–3968. https://doi.org/10.2147/DDDT.S120014

Sebile Azirak, Eyyup Rencuzogullari. The in vivo genotoxic effects of carvacrol and thymol in rat bone marrow cells. Environmental Toxicology, (24) 2008https://doi.org/10.1002/tox.20380.

Salma A. El-Marasy. Sally A. El Awdan.  Azza Hassan.  Heba M.I. Abdallah. Cardioprotective effect of thymol against adrenaline-induced myocardial injury in rats. Heliyon, Volume 6, Issue 7, e04431. DOI: 10.1016/j.heliyon.2020.e04431

Chamanara, M., Abdollahi, A., Rezayat, S.M. et al. Thymol reduces acetic acid-induced inflammatory response through inhibition of NF-kB signaling pathway in rat colon tissue. Inflammopharmacol 27, 1275–1283 (2019). https://doi.org/10.1007/s10787-019-00583-8

Mahran, Y., Badr, A.M., Aloyouni, S., Alkahtani, M.M., Sarawi, W.S., Ali, R., Alsultan, D., Almufadhili, S., Almasud, D.H., & Hasan, I.H. (Year). Morin hydrate protects against cisplatin-induced testicular toxicity by modulating ferroptosis and steroidogenesis genes’ expression and upregulating Nrf2/Heme oxygenase-1. Scientific Reports | (2025) 15:22720 | https://doi.org/10.1038/s41598-025-08235-4  

(Al-Khrashi et al., 2022) (Badr et al., 2025; Mahran et al., 2019)

Al-Khrashi, L.A., Badr, A.M., AL-Amin, M.A., Mahran, Y.F., 2022. Thymol ameliorates 5-fluorouracil-induced intestinal mucositis: Evidence of down-regulatory effect on TGF-β/MAPK pathways through NF-κB. J. Biochem. Mol. Toxicol. 36. https://doi.org/10.1002/jbt.22932

Badr, A.M., Aloyouni, S., Mahran, Y., Henidi, H., Elmongy, E.I., Alsharif, H.M., Almomen, A., Soliman, S., 2025. Thymol Preserves Spermatogenesis and Androgen Production in Cisplatin-Induced Testicular Toxicity by Modulating Ferritinophagy, Oxidative Stress, and the Keap1/Nrf2/HO-1 Pathway. Biomolecules 15, 1277. https://doi.org/10.3390/biom15091277

Mahran, Y.F., Badr, A.M., Aldosari, A., Bin-Zaid, R., Alotaibi, H.N., 2019. Carvacrol and Thymol Modulate the Cross-Talk between TNF-α and IGF-1 Signaling in Radiotherapy-Induced Ovarian Failure. Oxid. Med. Cell. Longev. 2019, 3173745. https://doi.org/10.1155/2019/3173745

  • Animal Husbandry and Euthanasia:
    Report precise environmental conditions (temperature, humidity, and light/dark cycle).
    Confirm compliance of the euthanasia method (CO₂ asphyxiation followed by decapitation) with AVMA 2020 guidelines.

Response:

Thank you for your helpful comment. The precise environmental conditions (temperature, humidity, and light/dark cycle) have now been reported in the revised Methods section. Additionally, we confirm that the euthanasia procedure (CO₂ asphyxiation followed by decapitation) was already clearly stated and confirmed in the manuscript. Revising the AVAM guidelines, our method was in accordance with them.

  • Sample Collection:

Clarify whether both testes were used for analysis and how samples were distributed across histological, biochemical, and molecular assays.

Verify the tissue storage temperatures—“37 °C” likely represents a typographical error (should be 4 °C).

Response:

Thank you for your insightful comment. We have now clarified that both testes were collected and that the samples were appropriately divided for histological, biochemical, and molecular analyses. This information has been added to the Methods section (line 148-158) for clarity.

Additionally, we confirm that the previously stated storage temperature of “37 °C” was a typographical error. The correct storage temperature is 4 °C, as corrected in the revised manuscript.

  • Biochemical and Molecular Assays:

Provide catalog numbers, manufacturers, and normalization methods for biochemical assays (e.g., per mg protein).

Indicate whether assays were performed in duplicate or triplicate.

For RT-qPCR, report primer efficiencies, annealing temperatures, and inclusion of negative controls.

For Western blotting, specify the gel percentage, blocking conditions, antibody sources/dilutions, and detection system.

Response:

Thank you for your detailed and constructive comments.

We have now provided the catalog numbers for all biochemical assay kits in the Methods section. Additionally, we clarified that all assays were performed in duplicate.

For the RT-qPCR experiments, primer efficiencies, annealing temperatures, and the use of no-template negative controls have been reported. Likewise, for Western blotting procedures, we have specified the gel percentages, blocking conditions, antibody sources and dilutions, and the detection system used. All of these details have been added to the revised manuscript to enhance methodological transparency.

  • Statistical Analysis:

Indicate whether data normality (Shapiro–Wilk test) and variance homogeneity (Levene’s test) were confirmed before ANOVA.

Clarify outlier handling, number of replicates, and exact significance thresholds (e.g., p < 0.05, p < 0.01).

Response:

Thank you for your insightful comment. We acknowledge the importance of confirming the assumptions underlying the statistical methods used. In our experiment, we used 8 biological replicates. We verified that the data followed a normal distribution using the Shapiro–Wilk test and confirmed the homogeneity of variances using Levene’s test before applying one-way ANOVA followed by Tukey’s post hoc test. We have now added a description of these preliminary tests in the "Statistical Analysis" section of the manuscript to clarify this point.

  • Formatting and Consistency:
    Standardize gene and protein nomenclature (e.g., SLC31A1, not SLC3A1).
    Ensure consistent formatting of units, degree symbols, and chemical notation (Cu²⁺).

Response:

Thank you for your observation. Done and corrected

  1. Results

Overall Evaluation

The results are logically structured and effectively demonstrate the protective effect of thymol against CP-induced testicular toxicity through oxidative, inflammatory, and cuproptotic mechanisms. However, numerical transparency, figure annotations, and methodological details can be improved.

Key Comments

  • Body and Testicular Weights:
    Define whether “body weight ratio” refers to testis/body-weight ratio or percentage body-weight change. Include numerical data (mean ± SEM) in the text or supplementary materials.

Response:

Thank you for your valuable comment. The numerical data were added in the text.

  • Sperm Parameters and Hormones:

Provide quantitative data for sperm morphology and viability percentages.

Specify the viability staining method used (e.g., eosin–nigrosin).

Indicate hormone concentration units and assay variability.

Add sample size (n) and exact p-values in figure legends.

Response:

Thank you for your insightful comment. Done and all required edit were completed and highlighted in revision manuscript

  • Oxidative Stress Markers:
    Clarify normalization (per mg protein or g tissue).
    Ensure consistency of units (e.g., nmol MDA/g tissue).
    Consider visual representation (bar graphs) in addition to tables.

Response:

Thank you for the valuable suggestions. We have clarified the normalization method and now consistently report the data as throughout the manuscript to ensure uniformity.

Additionally, we have included bar graphs instead of tables to provide clearer visual representation of the results, as recommended.

  • Inflammatory Markers (HMGB1, NF-κB, IL-8):
    Report ELISA results in consistent units.
    Clearly separate Western blot (protein expression) and ELISA (cytokine levels) data.
    Include representative blots labeled with molecular weights.

Response:

ELISA Units were based on the kit instructions. They were unified to pg/ml. representative blots labeled with molecular weights were added.

  • Cuproptosis Gene Expression:
    Provide fold-change values with SEM from 2^-ΔΔCt analysis.
    Confirm reference gene stability (β-actin).
    Ensure consistent gene nomenclature.
    Discuss correlations between cuproptosis and oxidative/inflammatory parameters.

Response:

Done.

The stability of the expression of the housekeeping genes actin-beta, Gapdh, HPRT, and UBC was analyzed throughout the different experimental groups. The result of the housekeeping genes expression stability revealed that the act-β was the most stably expressed gene since its mean value of the Cq and CT values indicated a non-significant difference across the different experimental groups which, clearly illustrated that the current treatments did not induce a significant effect on the expression level of the act-β across the different investigated groups that confirmed with the data that we obtained.

Gene nomenclature was unified.

The correlations between cuproptosis and oxidative/inflammatory parameters were discussed in the Discussion section.

  • Histopathology:
    Add semi-quantitative scoring (e.g., Johnsen’s score) to support descriptive findings.
    Include scale bars and consistent magnifications.
    Confirm that histological assessment was blinded.

Response:

Thank you for your valuable comment. Scale bars have been added to all histological images, and magnifications are now presented consistently across all figures. Furthermore, the histological assessment was performed by researchers who were blinded to the experimental groups.

For the semi-quantitative scoring, we already reported it in a previous study on the effect of thymol on cuproptosis-induced toxicity. As this study is mainly focused on exploring the mechanism of action, we used only the qualitative examination to ensure the model's success. This was referred to in the discussion part, and the reference was added.

Badr, A.M., Aloyouni, S., Mahran, Y., Henidi, H., Elmongy, E.I., Alsharif, H.M., Almomen, A., Soliman, S., 2025. Thymol Preserves Spermatogenesis and Androgen Production in Cisplatin-Induced Testicular Toxicity by Modulating Ferritinophagy, Oxidative Stress, and the Keap1/Nrf2/HO-1 Pathway. Biomolecules 15, 1277.

  1. Discussion and Conclusions

Overall Evaluation

The Discussion effectively integrates the study’s biochemical, molecular, and histological results, emphasizing the novel involvement of cuproptosis and the protective role of thymol. The logical flow is strong, but the section could be more concise, with clearer delineation between established knowledge and new contributions.

Major Comments

  • Contextualization:
    The initial discussion repeats background information already presented in the Introduction. Consider condensing to focus on new insights.
    Rephrase statements like “claims to minimize its use” to more neutral scientific language (e.g., “There is growing interest in adjunctive therapies that reduce cisplatin-induced toxicity…”).

Response:

Done.

  • Mechanistic Insights:

Expand on how CP triggers HMGB1 release (e.g., via DNA damage or mitochondrial dysfunction).
Clarify the sequence between oxidative stress and HMGB1/NF-κB activation.
Specify whether IL-8 elevation originates primarily in Leydig cells or seminiferous epithelium.
Discuss how CP may interfere with copper metabolism, thus activating cuproptosis.
Highlight novelty:

“This study provides the first evidence linking cuproptosis-related gene dysregulation to cisplatin-induced testicular injury.”

Response:

Done

  • Thymol Mechanisms:
    Summarize thymol’s multi-target effects:
    • Antioxidant:Increases SOD, reduces MDA.
    • Anti-inflammatory:Suppresses HMGB1/NF-κB/IL-8 signaling.
    • Anti-cuproptotic:Downregulates FDX1DLAT, and SLC31A1.
      Clarify whether these effects stem from direct copper modulation or secondary antioxidant activity.

Response:

Done

The discussion successfully synthesizes the biochemical, molecular, and histological findings, particularly highlighting the novel involvement of cuproptosis and the protective role of thymol in cisplatin-induced testicular injury. To further strengthen the contextual framework and emphasize the study’s novelty, I suggest incorporating recent relevant literature.

First, the study by Berman et et al. (2025), DOI: 10.1016/j.cbi.2025.111747 “Comprehensive characterization of poly(ADP-ribosyl)ation in spermatozoa as a novel and early biomarker of sperm health: A preliminary look,” provides valuable insights into early biomarkers of sperm health through poly(ADP-ribosyl)ation profiling. Including this reference would enrich the discussion of molecular markers linked to sperm integrity and damage induced by cisplatin, especially in relation to DNA damage and oxidative stress pathways.

Second, the recent work on nano Spirulina platensis (NSP) by Khalil et al. (2024), DOI: 10.1007/s00210-024-0131483-z “Nano Spirulina platensis countered cisplatin-induced repro-toxicity by reversing the expression of altered steroid hormones and downregulation of the StAR gene,” highlights another potential therapeutic avenue. NSP’s ability to restore steroid hormone balance and normalize StAR gene expression complements the current findings by emphasizing the importance of hormonal regulation and steroidogenesis in mitigating cisplatin toxicity. This adds a broader perspective on potential multi-target protective strategies that include antioxidant, anti-inflammatory, and endocrine modulatory effects.

Incorporating these references will:

  • Strengthen the argument that early molecular alterations (e.g., poly(ADP-ribosyl)ation) serve as sensitive indicators of sperm health, consistent with the observed HMGB1-mediated damage and oxidative stress.
  • Broaden the discussion on therapeutic interventions beyond thymol, pointing to complementary or synergistic strategies such as NSP for preserving fertility during chemotherapy.
  • Emphasize the multifactorial nature of cisplatin-induced testicular injury, involving oxidative stress, inflammation, steroidogenic disruption, and newly identified cell death pathways like cuproptosis.

Response:

Done.

Technical Corrections:

  • Use “H and E” consistently.
  • Correct citation formatting (e.g., “Kohsaka et al. (2020)” instead of “the finding of (Kohsaka et al. 2020)”).
  • Replace informal connectors (“Moreover”) with more formal alternatives (“Furthermore”).

Response:

Done

Summary:
The manuscript presents novel and significant findings on the involvement of HMGB1 and cuproptosis in CP-induced testicular toxicity and the protective role of thymol. To further strengthen the work, the authors should (1) refine transitions and mechanistic rationale in the Introduction, (2) enhance methodological transparency, (3) provide more detailed data presentation and figure labeling in the Results, and (4) focus the Discussion on mechanistic interpretation and study implications rather than repetition of results.

Reviewer 2 Report

Comments and Suggestions for Authors

This manuscript investigates cisplatin-induced testicular injury in rats and posits a mechanistic link among oxidative stress, HMGB1/NF-κB–driven inflammation, and cuproptosis (via SLC31A1/CTR1, FDX1, DLAT), while testing thymol as a protective agent. The study’s design spans hormones, oxidative stress markers, histology, yielding a coherent picture in which thymol attenuates inflammation and putative cuproptotic signaling. Strengths include a relevant in-vivo design and a reasonable alignment between phenotypes and the proposed pathways. However, several key elements need substantial improvement: (i) the Western blot does not meet standard reporting quality; (ii) protein-level validation of cuproptosis hallmarks is missing; (iii) figure callouts, dose consistency, and symbol usage need correction; and (iv) human anchoring (public TGCT datasets) is absent. Addressing these issues will be necessary for publication.

Major points

  1. To strengthen translational relevance, analyze cuproptosis pathway genes (e.g., FDX1, DLAT, LIAS, LIPT1, SLC31A1/CTR1, ATP7B, SCO1/2, LRPPRC, and Complex IV subunits) in testicular germ cell tumors using public data. These analyses are feasible with TCGA-TGCT  and published single-cell seminoma/TGCT data, it would substantively anchor your rat findings in human disease context.

  1. Protein-level validation of cuproptosis (beyond mRNA): Since mRNA levels do not always translate to protein abundance or modification state, please perform Western blot analyses—prioritizing DLAT—and, if feasible, detect high–molecular weight DLAT aggregates using non-reducing/Native PAGE to establish cuproptosis specificity.

  1. Figure 4D—The current HMGB1 Western lacks the clarity and rigor typically expected. Please repeat the HMGB1 Western with validated antibodies. For technical benchmarks, several groups working in the same or closely related cell systems have produced clear HMGB1 Westerns with proper controls

Minor points:

  1. Figure 1 (left) Body weight is misspelled
  2. Dose inconsistency (thymol). Abstract states 50 mg/kg p.o. daily ×2 weeks; Methods use 60 mg/kg. Please reconcile everywhere (including Fig/Methods text).
  3. Gene symbol and spelling errors (cuproptosis block).
  1. “SCL31A1” (Abstract) → SLC31A1.
  2. Section header uses “SLC3A1” (missing “1”)—should be SLC31A1; same in Fig. 5 caption.
  3. “FDx1” in Fig. 5 caption → FDX1.
  4. Multiple misspellings of cuproptosis (“cuprotosis,” “cuporoptosis”); fix globally.

5. Protein name spelling. “Ferrodoxin 1 (FDX1)” → Ferredoxin 1. Please correct throughout.

6. n per group inconsistency. Methods state 8 rats/group; several legends report n=6 (Fig. 1) and n=3 (Fig. 2). Please harmonize actual biological replicates, or explain why subsets differ

7. Figure callouts in Results. Results do not explicitly reference some panels ( “Fig. 2D–E”). Please add direct callouts with the direction of change so readers don’t miss baseline data.

Author Response

Response

Reviewer 2:

This manuscript investigates cisplatin-induced testicular injury in rats and posits a mechanistic link among oxidative stress, HMGB1/NF-κB–driven inflammation, and cuproptosis (via SLC31A1/CTR1, FDX1, DLAT), while testing thymol as a protective agent. The study’s design spans hormones, oxidative stress markers, histology, yielding a coherent picture in which thymol attenuates inflammation and putative cuproptotic signaling. Strengths include a relevant in-vivo design and a reasonable alignment between phenotypes and the proposed pathways. However, several key elements need substantial improvement: (i) the Western blot does not meet standard reporting quality; (ii) protein-level validation of cuproptosis hallmarks is missing; (iii) figure callouts, dose consistency, and symbol usage need correction; and (iv) human anchoring (public TGCT datasets) is absent. Addressing these issues will be necessary for publication.

 Major points

  1. To strengthen translational relevance, analyze cuproptosis pathway genes (e.g., FDX1, DLAT, LIAS, LIPT1, SLC31A1/CTR1, ATP7B, SCO1/2, LRPPRC, and Complex IV subunits) in testicular germ cell tumors using public data. These analyses are feasible with TCGA-TGCT and published single-cell seminoma/TGCT data; it would substantively anchor your rat findings in a human disease context.

 Response:

We strongly agree with the reviewer that anchoring our findings in a human disease context is crucial for translational relevance. We have performed the requested comprehensive analysis of key cuproptosis pathway genes in human testicular germ cell tumors (TGCT) using Bulk RNA-seq Analysis (TCGA-TGCT). We analyzed the expression levels of the core cuproptosis genes (FDX1, DLAT, SLC31A1), using the TCGA Testicular Germ Cell Tumors (TGCT) dataset using GEPIA2. Supplemental Figure 1, compares gene expression between TGCT tissues and adjacent normal tissues (or GTEx controls). The gene expression analysis of Cuproptosis-related genes —DLAT, FDX1, and SLC31A1 —using GEPIA 2 showed no significant difference between normal tissues (Match TCGA normal and GTEx data) and tumor tissues (TCGA-TCGT). This suggests the cuproptosis-mediated cell death is not the primary driver for testicular cancer, unlike other cancer types, or it might be stage-dependent.

These data were referred to in the manuscript.

  1. Protein-level validation of cuproptosis (beyond mRNA): Since mRNA levels do not always translate to protein abundance or modification state, please perform Western blot analyses—prioritizing DLAT—and, if feasible, detect high–molecular–weight DLAT aggregates using non-reducing/Native PAGE to establish cuproptosis specificity.

 Response:

We agree with the reviewer that mRNA alone cannot be sufficient. However, this is the first study to shed light on the possible role of cuproptosis in cisplatin-induced testicular damage. We also supported the data with the FDX1 protein level measured by ELISA and added Cu level data. DLAT is really considered as the final target that induces aggregate formation. However, FDX1 is also central to the process.

Furthermore, we hoped to measure DLAT protein expression, but due to Saudi Arabia's export regulations, it takes about 2 months to obtain a single antibody. We aim not to lose novelty. We added this point to the recommendations for future direction.

  1. Figure 4D—The current HMGB1 Western lacks the clarity and rigor typically expected. Please repeat the HMGB1 Western with validated antibodies. For technical benchmarks, several groups working in the same or closely related cell systems have produced clear HMGB1 Westerns with proper controls

 Response: Another HMGB1 western blot representitive was added. Thank you for this comment.

Minor points:

  1. Figure 1 (left) Body weight is misspelled

Response: Thank you for your observation. Done and corrected

  1. Dose inconsistency (thymol). Abstract states 50 mg/kg p.o. daily ×2 weeks; Methods use 60

Response: Thank you for your observation. Done and corrected

  1. mg/kg. Please reconcile everywhere (including Fig/Methods text).

Response: Thank you for your observation. Done and corrected

  1. Gene symbol and spelling errors (cuproptosis block).

Response: Thank you for your observation. Done and corrected

  1. “SCL31A1” (Abstract) → SLC31A1.

Response: Thank you for your observation. Done and corrected

  1. Section header uses “SLC3A1” (missing “1”)—should be SLC31A1; same in Fig. 5 caption.

Response: Thank you for your observation. Done and corrected

  1. “FDx1” in Fig. 5 caption → FDX1.

Response: Thank you for your observation. Done and corrected

  1. Multiple misspellings of cuproptosis (“cuprotosis,” “cuporoptosis”); fix globally.

Response: Thank you for your observation. Done and corrected

  1. Protein name spelling. “Ferrodoxin 1 (FDX1)” → Ferredoxin 1. Please correct throughout.

Response:Thank you for your observation. Done and corrected

  1. n per group inconsistency. Methods state 8 rats/group; several legends report n=6 (Fig. 1) and n=3 (Fig. 2). Please harmonize actual biological replicates, or explain why subsets differ

Response: Done. However, for Western Blotting, we used only 3 replicates. That is the cause for different n written.

  1. Figure callouts in Results. Results do not explicitly reference some panels ( “Fig. 2D–E”). Please add direct callouts with the direction of change so readers don’t miss baseline data.

Response: Done

Round 2

Reviewer 2 Report

Comments and Suggestions for Authors

The authors have sufficiently addressed my queries, and it can be published as such.